# Bioactive Polyketides from the Natural Complex of the Sea Urchin-Associated Fungi *Penicillium sajarovii* KMM 4718 and *Aspergillus protuberus* KMM 4747

**DOI:** 10.3390/ijms242316568

**Published:** 2023-11-21

**Authors:** Elena V. Leshchenko, Dmitrii V. Berdyshev, Ekaterina A. Yurchenko, Alexandr S. Antonov, Gleb V. Borkunov, Natalya N. Kirichuk, Viktoria E. Chausova, Anatoly I. Kalinovskiy, Roman S. Popov, Yuliya V. Khudyakova, Ekaterina A. Chingizova, Artur R. Chingizov, Marina P. Isaeva, Anton N. Yurchenko

**Affiliations:** 1G.B. Elyakov Pacific Institute of Bioorganic Chemistry, Far Eastern Branch of the Russian Academy of Sciences, 159 Prospect 100-Letiya Vladivostoka, Vladivostok 690022, Russia; berdyshev@piboc.dvo.ru (D.V.B.); antonov_as@piboc.dvo.ru (A.S.A.); gborkunov@gmail.com (G.V.B.); sheflera@bk.ru (N.N.K.); v.chausova@gmail.com (V.E.C.); kaaniv@piboc.dvo.ru (A.I.K.); popov_rs@piboc.dvo.ru (R.S.P.); 161070uuu@mail.ru (Y.V.K.); martyyas@mail.ru (E.A.C.); chingizov_ar@piboc.dvo.ru (A.R.C.); issaeva@piboc.dvo.ru (M.P.I.); yurchenkoan@piboc.dvo.ru (A.N.Y.); 2Institute of High Technologies and Advanced Materials, Far Eastern Federal University, Vladivostok 690922, Russia

**Keywords:** *Penicillium sajarovii*, *Aspergillus protuberus*, *ITS*, *beta-tubulin*, *calmodulin*, *RPB2*, phylogeny, polyketides, urease activity, antimicrobial activity, cardiomyocytes, infectious myocarditis

## Abstract

The marine-derived fungal strains KMM 4718 and KMM 4747 isolated from sea urchin *Scaphechinus mirabilis* as a natural fungal complex were identified as *Penicillium sajarovii* and *Aspergillus protuberus* based on Internal Transcribed Spacer (*ITS*), partial β-tubulin (*BenA*), and calmodulin (*CaM*) molecular markers as well as an ribosomal polymerase two, subunit two (*RPB2*) region for KMM 4747. From the ethyl acetate extract of the co-culture, two new polyketides, sajaroketides A (**1**) and B (**2**), together with (2′S)-7-hydroxy-2-(2′-hydroxypropyl)-5-methylchromone (**3**), altechromone A (**4**), norlichexanthone (**5**), griseoxanthone C (**6**), 1,3,5,6-tetrahydroxy-8-methylxanthone (**7**), griseofulvin (**8**), 6-O-desmethylgriseofulvin (**9**), dechlorogriseofulvin (**10**), and 5,6-dihydro-4-methyl-2H-pyran-2-one (**11**) were identified. The structures of the compounds were elucidated using spectroscopic analyses. The absolute configurations of the chiral centers of sajaroketides A and B were determined using time-dependent density functional theory (TDDFT)-based calculations of the Electronic Circular Dichroism (ECD) spectra. The inhibitory effects of these compounds on urease activity and the growth of *Staphylococcus aureus*, *Escherichia coli*, and *Candida albicans* were observed. Sajaroketide A, altechromone A, and griseofulvin showed significant cardioprotective effects in an in vitro model of *S. aureus*-induced infectious myocarditis.

## 1. Introduction

Marine microbial ecosystems are characterized by an uneven ratio between prokaryotes and eukaryotes. For example, an investigation of the microbial community during a natural *Noctiluca scintillans* algal bloom in the coastal area of Dongchong (Shenzhen, China) showed that, in addition to dinoflagellates, Bacteria and Archaea predominate in this community, and Fungi are represented by only four phyla in very small quantities [1]. This leads to considerable competition within the community and the production of exolites to control the competitors. A number of reports have confirmed that the produced secondary metabolites may be extensively involved in a variety of communication events among microorganisms when a bacteria–fungus co-culture produces bacteriostatic and fungicidal exolites [2]. From 2009 to 2019, various alkaloids, anthraquinones, cyclopeptides, macrolides, phenylpropanoids, polyketides, steroids, terpenoids, and others (153 compounds in total) with antimicrobial, cytotoxic, hemolytic, and anti-proliferative activities were isolated from marine microbial co-cultures [3]. The latest report by Li et al. described 194 compounds with cytotoxic, antibacterial, antifungal, antimalarial, and antifouling properties isolated from marine microbial co-cultures in 2012–2022 [4]. Our study of various marine fungus–fungus co-cultures resulted in the isolation of new cytotoxic and hemolytic metabolites [5,6,7].

In the course of our investigation of the fungal community of various marine substrates in the Sea of Japan, a natural fungal complex was isolated from the aboral surface of the sea urchin *Scaphechinus mirabilis* collected from Troitsa Bay and formed by only two strains, KMM 4718 and KMM 4747. The fungal species *Penicillium sajarovii* is one of three representatives of the *Raistrickiorum* series (*Ramosa*) and is phylogenetically close to the species *P. raistrickii* [8]. Various compounds have been reported to be isolated from *P. raistrickii*. Among them, there are benzo-fused 2,8-dioxabicyclo[3.3.1]nonane-contained spiroketals [9]; several *p*-terphenyl and xanthone derivatives together with griseofulvin-related metabolites [10]; indole diketopiperazine alkaloids and benzodiazepine derivatives [11]; radical scavenging phenetyltetrahydrofuranes [12]; and cytotoxic spiroditetrahydropyran polyketides [13]. The fungal species *Aspergillus protuberus* belongs to the *Versicolores* series (*Nidulantes*), which currently includes 17 species, and is phylogenetically close to the species *A. versicolor*, *A. amoenus*, *A. tabacinus*, and *A. austroafricanus* [8]. These fungi are well-known sources of polyketides, diketopiperazines, sterols, terpenoids, and meroterpenoids [14,15,16]. *Penicillium* and *Aspergillus* species are two of the most widespread fungal organisms on Earth, growing on many different substrates, and are permanent components of marine microbial communities [17,18]. Each of these studied fungi is a promising producer of various metabolites with a wide range of biological activities; therefore, studying their natural complex can yield interesting results.

Here, we report the isolation and structure elucidation, including absolute configuration determination, of 11 polyketides (Figure 1) from the natural complex of the sea urchin-associated fungi *P. sajarovii* KMM 4718 and *A. protuberus* KMM 4747. The antimicrobial and cytotoxic activities of the isolated compounds are also evaluated.

## 2. Results

### 2.1. Molecular Identification of the Fungal Strains

In this study, to clarify the taxonomic position of the strains KMM 4718 and KMM 4747, we sequenced molecular markers, such as *ITS*, partial *BenA*, and *CaM* regions. Approximately 550–570 bp fragments of the *ITS* region, about 430–450 bp fragments of partial *BenA*, and about 500–520 bp fragments of the *CaM* gene were successfully amplified. In addition, a 940 bp fragment of the *RPB2* gene was amplified for KMM 4747. A BLAST search showed that the *ITS* region, partial *BenA*, and *CaM* gene sequences of strain KMM 4718 were 100% identical to those of the ex-type strain *Penicillium sajarovii* CBS 277.83^T^. The phylogenetic ML tree of the concatenated *ITS*-*BenA*-*CaM* gene sequences clearly showed that the strain KMM 4718 clustered with ex-type strain *Penicillium sajarovii* CBS 277.83^T^ (Figure 2).

A BLAST search showed that the *ITS* region was 100% identical to the sequence of the ex-type strain *Aspergillus protuberus* CBS 602.74^T^, whereas the partial *BenA*, *CaM*, and *RPB2* gene sequences were more than 99% identical. The phylogenetic ML tree of the concatenated *ITS*-*BenA*-*CaM*-*RPB2* gene sequences clearly showed that the strain KMM 4747 clustered with ex-type strain *Aspergillus protuberus* CBS 602.74^T^ (Figure 3).

### 2.2. Structure Elucidation

The molecular formula of **1** was established to be C_12_H_12_O_5_ by HRESIMS (*m*/*z* 259.0576 [M + Na]^+^) (Appendix A) and was confirmed by ^13^C NMR analysis. A close inspection of ^1^H, ^13^C NMR (Table 1, Appendix A), DEPT, and HSQC data of **1** revealed the presence of two methyl groups (δ_H_ 1.60, δ_C_ 35.2), including one oxygen-bearing (δ_H_ 3.89, δ_C_ 56.9), three *sp^2^* methines (δ_H_ 5.52, δ_C_ 99.6; δ_H_ 6.25, δ_C_ 102.4; δ_H_ 6.77, δ_C_ 106.1), an oxygenated quaternary *sp^3^*-carbon (δ_C_ 70.7), and five quaternary *sp^2^*-carbons (δ_C_ 107.5; δ_C_ 151.5), including four oxygen-bearing ones (δ_C_ 164.1, 164.4, 179.3, and 190.3). These data and seven degrees of unsaturation from the molecular formula suggest that compound **1** possesses two rings, including one aromatic system, one double bond, and one carbonyl. The HMBC correlations (Appendix A) from H-5 (δ_H_ 6.77) to C-4 (δ_C_ 70.7), C-7 (δ_C_ 102.4), C-8a (δ_C_ 107.5), and C-6 (δ_C_ 164.1); from H-7 (δ_H_ 6.25) to C-5 (δ_C_ 106.1) and C-8a; and from H-2 (δ_H_ 5.52) to C-4, C-8a, C-3 (δ_C_ 179.3), and C-1 (δ_C_ 190.3) revealed the structure of the naphthalenone framework with a Δ^2^ double bond and hydroxy groups at C-6 and C-8 in the ring A. The HMBC correlations from H_3_-9 (δ_H_ 1.60) to C-4, C-4a (δ_C_ 151.5), and C-3 established the location of the C-9 methyl group at C-4. The location of the methoxy group at C-3 was established based on the HMBC correlation from 3-OMe (δ_H_ 3.89) to C-3. The downfield chemical shift and remaining functionality of C-3 suggested an additional OH group at C-3. Thus, the planar structure of **1** was determined, and this compound was named sajaroketide A. It should be noted that sajaroketide A (**1**) is a derivative of the well-known fungal pentaketide flaviolin [19].

The molecular formula of **2** was established to be C_13_H_14_O_5_ by HRESIMS (*m*/*z* 273.0735 [M + Na]^+^) (Appendix A) and was confirmed by the ^13^C NMR spectrum. The ^1^H and ^13^C NMR data (Table 1, Appendix A) observed for compound **2** closely resembled those obtained for **1,** except for the additional methoxy group (δ_H_ 3.87, δ_C_ 55.9). The location of this group at C-6 was established based on the HMBC correlation from 6-OMe to C-6 (δ_C_ 166.1). Thus, the planar structure of **2** was determined to be the 6-O-methyl derivative of sajaroketide A, and this compound was named sajaroketide B.

Unfortunately, correlations in the ROESY spectrum were not informative and could not be used to establish the relative configurations of the stereocenters of **1** and **2**. Thus, theoretical and experimental ECD spectra were compared to determine the absolute configurations of the chiral centers of sajaroketides A (**1**) and B (**2**). UV and ECD spectra were calculated for the most stable conformations of **1** and **2** using the time-dependent density functional theory (TDDFT_B3LYP). All calculations were performed with the Gaussian 16 suite of programs [20] using ultra-fine integration grids and very tight optimization convergence criteria.

The performed preliminary modeling showed that the signs and intensities of individual bands in their ECD spectra strongly depended on the large-amplitude motions (LAMs) occurring in these compounds—the internal rotations of hydroxy and methoxy groups around C(3)-O(3), C(4)-O(4), and C(6)-O(6) bonds, the tautomeric rearrangement, and the inversion-type motion of cycle “A”. For λ ≤ 240 nm, these dependencies complicated the task of the ECD spectral interpretation. The rigorous accounting of contributions from these LAMs to spectral properties must be performed based on some kind of intramolecular dynamics modeling—the Schrödinger equations for the motions along these LAMs’ degrees of freedom must be constructed and solved. This approach is far from the present standard theoretical scheme, which is generally used in a stereochemical analysis. An alternative to this is the approach where efforts are focused on the investigation of those parts of ECD spectra where individual bands are well separated and the influence of LAMs may be accounted for qualitatively and correctly, even based on the standard theoretical scheme with moderate improvements. The latter approach was used in this study, and the characteristic energy region chosen for the investigation was λ ≥ 230 nm. Calculations, performed at the “PCM level”, showed that conformation R-1_c1 was the most thermodynamically stable—its Gibbs free energy was minimal (Appendix A). The conformation **4*R*-1_c2** is less stable then **4*R*-1_c1** for about ΔG ≈ 0.45 kcal/mol. Figure 4 and Figure 5 demonstrate the UV and ECD spectra, respectively, for these two most stable conformations of 4*R*-**1**, calculated using TDDFT_B3LYP/cc-pvTz_PCM//B3LYP/cc-pvTz_PCM methods at the “PCM level” of theory.

While the UV spectra of **4*R*-1_c1** and **4*R*-1_c2** are nearly similar, their ECD spectra differ qualitatively in the 220 ≤ λ ≤ 270 nm region. The UV shift, obtained from a comparison of the theoretical statistically averaged and experimental UV spectra, was Δλ ≈ +7 nm. This UV shift value was used to construct the average ECD spectrum (Figure 5).

The theoretically averaged ECD spectrum of 4R-1 satisfactorily reproduced the experimental ECD spectrum of 1, except for the 200 ≤ λ ≤ 250 nm region. At the same time, the ECD spectrum of the “minor” conformer **4*R*-1_c2** was in very good agreement with the experiment at λ ≥ 230 nm. This may provide evidence that the PCM approach and standard theoretical scheme alone fail to account for the full measure of the compound-solvent interaction, and hence to correctly describe the thermodynamics of the conformational rearrangement process. To overcome these discrepancies, we used a more complex theoretical model, in which the direct modeling of sajaroketide A (**1)** interacting with two methanol molecules (4*R*-**1**&(CH_3_OH) × 2 and 4*R*-**2**&(CH_3_OH) × 2) was performed. The different variants of the intermolecular hydrogen-bond formation (IMHB) were modeled using the B3LYP/cc-pvTz_PCM approach. The optimized structures of the seven most stable conformations are shown in Appendix A.

We found that the thermodynamic equilibrium in cluster 4*R*-**1**&(CH_3_OH) × 2 was dislocated to the conformations of the **4*R*-1_c2** type (structures Direct_*R*-1_c–Direct_*R*-1_c5), and this prevalence governed the existence in the experimental ECD spectrum of the negative band at λ ≈ 230–240 nm. The relative intensities of the positive bands in the λ ≥ 255 nm diapason were also reproduced properly at the “direct level” than at the “PCM level” of theory (Figure 6). Therefore, the absolute structure of **1** is 4*R* (for demonstration, the theoretical ECD spectrum of 4*S*-**1** is plotted in Figure 6 using a dashed line).

The experimental ECD and UV spectra of **2** were similar to those of 4*R*-**1**. A comparison of the theoretical ECD spectra of 4*R*-**2** and 4*S*-**2** calculated at the “direct level” of theory with the experimental one is shown in Figure 7. Spectra Δε_exp_ and Δε_calc_(4*R*-**2**) are in good qualitative agreement, while Δε_calc_(4*S*-**2**) “fails” to reproduce experimental data throughout the diapason. Thus, the absolute structure of **2** is 4*R* (for demonstration, the theoretical ECD spectrum of 4*S*-**2** is plotted in Figure 6 using a dashed line).

The molecular formula of compound **3** was determined to be C_13_H_14_O_4_ by HRESIMS (*m*/*z* 257.0788 [M + Na]^+^) (Appendix A), which corresponded to seven degrees of unsaturation. These data and a careful inspection of the ^13^C and ^1^H NMR spectra (Appendix A), including HSQC, HMBC, COSY, and DEPT experiments, as well as a comparison with the literature data, identify compound **3** as 7-hydroxy-2-(2′-hydroxypropyl)-5-methylchromone, a known metabolite of *Penicillium solitum* and *P. griseofulvum* [21,22,23]. A comparison of the specific optical rotation values of [α]_D_^20^ + 21.3 (for **3**) and [α]_D_^21^ + 38.4 (for 7-hydroxy-2-(2′*S*-hydroxypropyl)-5-methylchromone in the literature data [22,24]) proved the absolute configuration of the stereo center in **3** as 2′*S*.

The structures of other isolated compounds were identified based on HRESIMS and NMR data with known fungal metabolites: altechromone A (**4**) [25], norlichexanthone (=fusarindin) (**5**) [26], griseoxanthone C (**6**) [27], 1,3,5,6-tetrahydroxy-8-methylxanthone (**7**) [10], griseofulvin (**8**), 6-O-desmethylgriseofulvin (**9**), dechlorogriseofulvin (**10**) [28], and 5,6-dihydro-4-methyl-2H-pyran-2-one (3-methyl-2-penten-5-olide) (**11**) [21].

### 2.3. Bioactivity of Isolated Compounds

#### 2.3.1. Antimicrobial Activity

The effects of compounds **1**–**9** and **11** on urease enzyme activity as well as the growth of *Staphylococcus aureus*, *Escherichia coli*, and *Candida albicans* strains were tested, and the data are presented in Table 2. Compound **10** was not tested because its quantity was insufficient.

Urease is an important target for the treatment of urease-related bacterial infections [29]. In this study, several fungal metabolites showed a significant inhibition of urease activity in the cell-free test. So, the half-maximal inhibition of urease activity by norlichexanthone (**5**), griseofulvin (**8**), and 5,6-dihydro-4-methyl-2H-pyran-2-one (**11**) was observed at concentrations of 15.3, 10.1, and 11.4 µM, respectively (Table 2). New sajaroketide B (**2**), griseoxanthone C (**6**), and 1,3,5,6-tetrahydroxy-8-methylxanthone (**7**) inhibited urease activity by 50% at concentrations of 98.5, 97.3, and 100 µM, respectively. Compounds **1**, **3**, **4**, and **9** had no effect on urease activity at concentrations of up to 100 µM.

New sajaroketide A (**1**) at 100 µM inhibited *S. aureus* and *E. coli* growth by 35.1% and 25.4%, respectively, whereas sajaroketide B (**2**) was inactive (Table 2). Chromone derivative **3**, altechromone A (**4**), norlichexanthone (**5**), and griseofulvin (**8**) at concentrations of 100 µM inhibited *S. aureus* growth by 17.7%, 30.1%, 20.8%, and 11.5%, respectively. Moreover, compounds **4**, **5**, and **8** inhibited *E. coli* growth by 13.6%, 27.9%, and 16.8%, respectively. 6-O-Desmethylgriseofulvin (**9**) inhibited both *S. aureus* and *E. coli* growth by nearly 23%. Xanthon-related polyketide **7** at 100 µM inhibited only *C. albicans* growth by 15.4%. Compounds **6** and **11** did not affect the growth of all test strains.

#### 2.3.2. Cytotoxic Activity

The cytotoxic activities of compounds **1**–**9** and **11** toward human hepatocarcinoma HepG2 and normal rat cardiomyocyte H9c2 cells were evaluated. Cell viability and the half-maximal concentration of the cytotoxic effects are presented in Table 3.

Sajaroketide A (**1**) at 100 µM decreased H9c2 cell viability by 34.6%. Sajaroketide B (**2**) at 100 µM decreased HepG2 and H9c2 cell viability values by 23.7% and 21.0%, respectively. Nevertheless, sajaroketides A (**1**) and B (**2**) were nontoxic to both cell lines at a concentration of 10 µM.

Chromone derivative **3** was non-toxic to HepG2 cells and decreased H9c2 cell viability by 25.7% at 100 µM. Altechromone A (**4**) was non-toxic to both cell lines at concentrations up to 100 µM.

Norlichexanthone (**5**) at 100 µM decreased HepG2 and H9c2 cell viability values by 47.4% and 51.0%, respectively; therefore, the IC_50_ was nearly 100 µM for both cell lines. Griseoxanthone C (**6**) at 100 µM decreased HepG2 and H9c2 cell viability values by 72.9% and 36.7%, respectively, and its IC_50_ was calculated as 64.7 µM for HepG2 cells and higher than 100 µM for H9c2 cells. Polyketide **7** at 100 µM decreased HepG2 and H9c2 cell viability values by 42.6 and 47.1 µM, respectively, so the IC_50_ was nearly 100 µM for both cell lines. Griseofulvin (**8**) at 100 µM diminished HepG2 cell viability by 60.1% and the IC_50_ was calculated as 77.8 µM. However, griseofulvin (**8**) at 100 µM decreased H9c2 cell viability only by 28.4%. Compound **11** at 100 µM affected HepG2 cell viability by 23.4% and did not affect H9c2 cell viability.

#### 2.3.3. The Effect of Compounds against *S. aureus* Infection Damage of H9c2 

The following step of the study was to determine the cytoprotective effects of compounds **1**–**9** and **11** in an in vitro model of infectious myocarditis when H9c2 cardiomyocytes were co-cultured with an *S. aureus* suspension. We treated *S. aureus*-infected H9c2 cells with the compounds **1**–**9** and **11** at concentrations of 1 and 10 µM and measured the viability of the cells by MTT as described in the Section 4 (Figure 8). 

New sajaroketide A (**1**) increased the viability of *S. aureus*-infected H9c2 cardiomyocytes by 50.8% and 49.2% at 1 and 10 µM, respectively. Chromone derivative **3** at 10 µM increased the viability of these cells by 24.9%. Altechromone A (**4**) increased the viability of these cells by 17.4% and 39.9% at 1 and 10 µM, respectively. Norlichexanthone (**5**), xanthone derivative **7**, griseofulvin (**8**), and 6-O-desmethylgriseofulvin (**9**) at 10 µM increased the viability of these cells by 23.9%, 33.9%, 47.7%, and 28.9%, respectively. Compounds **2**, **6**, and **11** did not show any statistical effect on the viability of *S. aureus*-infected H9c2 cardiomyocytes.

## 3. Discussion

Thus, 11 polyketides, including two new sajaroketides, A and B, were isolated from the natural complex of *P. sajarovii* KMM 4718 and *A. protuberus* KMM 4747 associated with sea urchin *S. mirabilis*.

The anti-infection effect of norlichexanthone (**5**) and griseofulvin (**8**) correlated with their influence on *S. aureus* growth and urease activity. Chromone derivative **3** and griseofulvin-related compound **9** did not show any effect on urease activity, but inhibited *S. aureus* growth and, so may have inhibited bacterial growth in the in vitro model of myocarditis. Xanthone derivative **7** did not show any effect on *S. aureus* growth, but inhibited the activity of urease, which might be of relevance to this case.

Griseofulvin is a known antifungal polyketide produced by various fungi [30]. It was approved by the Food and Drug Administration (FDA) in 1959 as an anti-dermatophyte drug with an anti-inflammatory effect, as well as improving coronary blood flow and decreasing blood pressure [31]. Moreover, the inhibition of tumor growth and several forms of cancer cell proliferation were found [32]. Griseofulvin, currently used per os for the treatment of scalp dermatophytosis, achieving smooth skin, and improving the nails, and our new data concerning its possibility to inhibit urease activity and its cardioprotective effect on in vitro infectious myocarditis are interesting subjects.

Sajaroketide A (**1**) and altechromone A (**4**) showed greater cytoprotective effects in these experiments. It has previously been published in the literature that altechromone A (**4**) was substantially active against *Bacillus subtilis*, *Escherichia coli*, *Pseudomonas fluorescens*, and *Candida albicans* with the MICs of 3.9, 3.9, 1.8, and 3.9 μg/mL, respectively [33]. In our experiments, altechromone A (**4**) not only inhibited the growth of *S. aureus*, but was also effective against the staphylococcal infection of cardiomyocytes H9c2. Sajaroketide A (**1**) inhibited the growth of *S. aureus* similar to altechromone A (**4**), but its effect on *S. aureus*-infected H9c2 cells was more significant at a concentration of 1 µM. The influences of **1** and **4** on urease activity were very poor, which indicated their effects on *S. aureus* defined by other mechanisms.

According to the Global Burden of Disease project, in 2019, 0.428 million cases of endocarditis were recorded worldwide and 0.0663 million of them led to the deaths of patients. Over 10 years (2010–2019), the number of cases increased by 29% and the number of deaths by 21.8%. In 2019, 4.06 million cases of cardiomyopathy and myocarditis of various origins were recorded worldwide, and almost 10% (0.34 million) of the cases resulted in the deaths of patients. Over 10 years (2010–2019), the number of cases increased by 20% and the number of deaths by 3.3%. It should be noted that a significant proportion (30%) of myocarditis was caused by alcohol intoxication, but drug damage and infectious effects (viral and bacterial) played a dominant role [34]. Thus, the discovery of new compounds effective against staphylococcal infections is a potential future study of sajaroketide A.

## 4. Materials and Methods

### 4.1. General Experimental Procedures

Optical rotations were measured on a PerkinElmer 343 polarimeter (PerkinElmer, Waltham, MA, USA) in MeOH. UV spectra were recorded on a Shimadzu UV-1601PC spectrometer (Shimadzu Corporation, Kyoto, Japan) in MeOH. ECD spectra were measured using a Chirascan-Plus CD Spectrometer (Leatherhead, UK) in MeOH. ^1^H and ^13^C NMR spectra were recorded in aceton-d_6_ on Bruker Avance-500 and Avance III-700 spectrometers (Bruker BioSpin GmbH, Rheinstetten, Germany) operating at 500 and 125 MHz and 700 and 176 MHz, respectively, using TMS as an internal standard. HRESIMS spectra were obtained using a Bruker maXis Impact II mass spectrometer (Bruker Daltonics GmbH, Rheinstetten, Germany). 

Low-pressure liquid column chromatography was performed using Si gel KSK (50/100 μm, Imid Ltd., Krasnodar, Russia) and Gel ODS-A (12 nm, S—75 um, YMC Co., Ishikawa, Japan). Plates precoated with Si gel (5–17 μm, 4.5 × 6.0 cm, Imid) and Si gel60 RP-18 F254S (20 × 20 cm, Merck KGaA, Darmstadt, Germany) were used for thin-layer chromatography. Preparative HPLC was performed on Shimadzu LC-20 (Shimadzu, Kyoto, Japan) and Agilent 1100 (Agilent Technologies, Santa Clara, CA, USA) chromatographs using Shimadzu RID-20A (Shimadzu, Kyoto, Japan) and Agilent 1100 (Agilent Technologies, Santa Clara, CA, USA) refractometers and YMC ODS-AM (YMC Co., Ishikawa, Japan 5 μm, 250 × 10 mm), Synergi, Fusion-RP (Phenomenex, Torrance, CA, USA, 4 μm, 250 × 10 mm), Synergi, Hydro-RP (Phenomenex, 4 μm, 250 × 10 mm), and HyperClone ODS (Phenomenex, Torrance, CA, USA, 5 μm, 250 × 4.6 mm) columns.

### 4.2. Fungal Strains

The fungal culture used in this study was likely a natural fungal complex isolated from the aboral surface of the sea urchin S. mirabilis collected from the Sea of Japan (Troitsa bay). This complex is a co-culture of the filamentous fungi *P. sajarovii* and *A. protuberus*.

Initially, in the course of studying the metabolic profile of the fungal isolate, an isolate was selected that visually presented signs of a monoculture of *P. sajarovii*, both when growing on agar media for analytical cultivation (wort agar) and diagnostics (Czapek’s medium with yeast extract), and during further preparative cultivation on a medium with rice. During the microscopy of the culture, fragments of mycelium were found that did not belong to *P. sajarovii* and indicated the presence of a co-culture of a fungus of an unknown taxonomic affiliation, which was confirmed by a molecular genetic analysis. Subsequently, the components of the fungal complex were dispersed and their monocultures were obtained. The molecular genetic analysis of the second component of the fungal complex showed that it belonged to the species *A. protuberus*.

The resulting fungal strains were stored in the Collection of Marine Microorganisms (PIBOC FEB RAS, Vladivostok, Russia) as *P. sajarovii* KMM 4718 and *A. protuberus* KMM 4747.

### 4.3. DNA Extraction and Amplification

Genomic DNA were isolated from fungal mycelia (mycelium) grown on MEA (malt extract agar) at 25 °C for 7 days, using the MagJET Plant Genomic DNA Kit (Thermo Fisher Scientific, Waltham, MA, USA), according to the manufacturer’s protocol. PCR was conducted using GoTaq Flexi DNA Polymerase (Promega, Madison, WI, USA). For the amplification of the *ITS* regions, the standard primer pair *ITS*1 and *ITS*4 was used [35]. The reaction profile was an initial denaturation at 95 °C for 300 s, followed by 10 cycles of 95 °C for 20 s, 52 °C for 30 s, and 72 °C for 90 s; then 25 cycles of 95 °C for 20 s, 55 °C for 30 s, and 72 °C for 90 s; and finally 72 °C for 300 s. For the amplification of the partial *BenA* genes, the standard primer pair Bt-2a and Bt-2b were used [36]. The reaction profile was 95 °C for 300 s, 35 cycles of 95 °C for 20 s, 60 °C for 20 s, and 72 °C for 90 s, and finally 72 °C for 300 s. For the amplification of the partial *CaM* genes, the degenerate primer pair cal_P/A_F (5′-TCYGAGTACAAGGAGGCSTT-3′) and cal_P/A_R (5′-CCRATGGAGGTCATRACGTG-3′) were used. For the strain KMM 4718, the reaction profile was an initial denaturation at 95 °C for 300 s, followed by 10 cycles of 95 °C for 20 s, 60 °C for 30 s, and 72 °C for 90 s; then 25 cycles of 95 °C for 20 s, 65 °C for 30 s, and 72 °C for 90 s; and finally 72 °C for 300 s. For the strain KMM 4747, the reaction profile was 95 °C for 300 s, 35 cycles of 95 °C for 20 s, 65 °C for 30 s, and 72 °C for 90 s, and finally 72 °C for 300 s. For the amplification of the partial *RPB2* gene of the strain KMM 4747, the degenerate primer pair rpb2_Asp_For (5′-ACCCGWGTCACCCGTGAYCTTCA-3′) and rpb2_Asp_R (5′-TACTYGGRTGRATCTCGCAGT-3′) were used. The reaction profile was an initial denaturation at 95 °C for 300 s, followed by 10 cycles of 95 °C for 20 s, 55 °C for 30 s, and 72 °C for 90 s; then 25 cycles of 95 °C for 20 s, 60 °C for 30 s, and 72 °C for 90 s; and finally 72 °C for 420 s. The amplified *ITS*, *BenA*, *CaM*, and *RPB2* genes were purified with the ExoSAP-IT™ PCR Product Cleanup Reagent (Thermo Fisher Scientific, Waltham, MA, USA). Sequencing was bidirectionally performed with the same primers on an Applied Biosystems SeqStudio Genetic Analyzer (Thermo Fisher Scientific, Waltham, MA, USA) using the Big Dye Terminator reagent kit, version 3.1. The gene sequences were deposited in GenBank under accession numbers OR431843 (KMM 4718) and OR431842 (KMM 4747) for *ITS*, OQ466617 (KMM 4718) and OQ466615 (KMM 4747) for partial *BenA*, OR451997 (KMM 4718) and OR451996 (KMM 4747) for partial *CaM*, and OR451998 (KMM 4747) for partial *RPB2* (Table 4). 

### 4.4. Phylogenetic Analysis

The *ITS* region, partial *BenA* and *CaM* gene sequences, fungal strain KMM 4718, and members of genus *Penicillium* (*Ramosum*), series *Lanosa*, *Raistrickiorum*, *Scabrosa*, *Soppiorum*, and *Virgata* were aligned by MEGA X software version 11.0.9 [37] using the Clustal W algorithm. The ex-type homologs were searched in the GenBank database (http://ncbi.nlm.nih.gov) using the BLASTN algorithm (http://www.ncbi.nlm.nih.gov/BLAST, accessed on 20 July 2023). The phylogenetic analysis was conducted using MEGA X software [37]. The *ITS* region and partial *BenA* and *CaM* gene sequences were concatenated into one alignment. A phylogenetic tree was constructed according to the maximum likelihood (ML) algorithm based on a general time-reversible model [38]. The tree topology was evaluated by 1000 bootstrap replicates. The *Talaromyces marneffei* CBS 388.87 ^T^ strain was used in the phylogenetic analysis as an outgroup (Table 4).

The *ITS* region, partial *BenA*, *CaM*, and *RPB2* gene sequences, fungal strain KMM 4747, and members of genus *Aspergillus*, series *Versicolores*, were aligned by MEGA X software version 11.0.9 [37] using the Clustal W algorithm. The search for the ex-type homologs and phylogenetic analysis were performed as described above. The *ITS* region and partial *BenA*, *CaM*, and *RPB2* gene sequences were concatenated into one alignment. A phylogenetic tree was constructed according to the maximum likelihood algorithm based on the Kimura 2-parameter model [39]. The tree topology was evaluated by 1000 bootstrap replicates. The *Talaromyces marneffei* CBS 388.87^T^ strain was used in the phylogenetic analysis as the outgroup (Table 4).

### 4.5. Cultivation of Penicillium sajarovii KMM 4718 and Aspergillus protuberus KMM 4747

The fungi were grown stationary at 22 °C for 21 days in 100 Erlenmeyer flasks (500 mL), each containing 20 g of rice, 20 mg of yeast extract, 10 mg of KH_2_PO_4_, and 40 mL of natural sea water (Marine Experimental Station of G.B. Elyakov Pacific Institute of Bioorganic Chemistry, Troitsa (Trinity) Bay, Sea of Japan). 

### 4.6. Extraction and Isolation 

At the end of the incubation period, the mycelia and medium were homogenized and extracted with EtOAc (3 L). The obtained extract was dried in vacuo. The residue was dissolved in H_2_O−EtOH (4:1) (500 mL) and was extracted with *n*-hexane (0.3 L × 4) and EtOAc (0.3 L × 10). After the evaporation of the EtOAc layer, the residual material (60.182 g) was passed over a silica column (4 × 55 cm), which was eluted followed by a step gradient from 100% *n*-hexane to EtOAc (total volume: 50 L). Fractions of 250 mL were collected and combined based on the TLC results (Si gel, toluene–isopropanol, 6:1 and 3:1). As a result, 2 fractions were obtained: Pr-4718-6 (1.735 g) and Pr-4718-9 (4.398 g). 

The *n*-hexane–EtOAc (80:20) fraction (Pr-4718-6, 1.735 g) was purified on a Gel ODS-A column eluted with MeOH–H_2_O (80:20) to yield the subfraction Pr-4718-6-80 (1.320 g), which was purified on a YMC-ODS-AM column eluted with CH_3_CN−H_2_O (60:40) to yield subfractions Pr-4718-6-80-1 (126 mg) and Pr-4718-6-80-3 (89 mg). Subfraction Pr-4718-6-80-1 (126 mg) was purified on a Hydro column eluted with CH_3_CN−H_2_O (45:55) to yield **2** (5.9 mg), **4** (2.4 mg), and **11** (1.5 mg). Subfraction Pr-4718-6-80-3 (89 mg) was purified on a Hydro column eluted with CH_3_CN−H_2_O (80:20) to yield subfractions Pr-4718-6-80-3-I (19 mg) and Pr-4718-6-80-3-II (30 mg). Subfraction Pr-4718-6-80-3-I (19 mg) was purified on a HyperClone column eluted with CH_3_CN−H_2_O (50:50) to yield **5** (8.2 mg). Subfraction Pr-4718-6-80-3-II (30 mg) was purified on a HyperClone column eluted with CH_3_CN−H_2_O (60:40) to yield **6** (7.9 mg). 

The *n*-hexane–EtOAc (80:20) fraction (Pr-4718-9, 4.398 g) was purified on a Gel ODS-A column eluted with MeOH–H_2_O (80:20) to yield the subfraction Pr-4718-9-80 (2.118 g), which was purified on a YMC-ODS-AM column eluted with CH_3_CN−H_2_O (60:40) to yield subfraction Pr-4718-9-80-1 (239.7 mg). Subfraction Pr-4718-9-80-1 (239.7 mg) was purified on a Hydro column eluted with CH_3_CN−H_2_O (40:60) to yield subfractions Pr-4718-9-80-1-I (164.3 mg), Pr-4718-9-80-1-2 (4.9 mg), Pr-4718-9-80-1-3 (11.5 mg), and Pr-4718-9-80-1-10 (4.4 mg). Subfraction Pr-4718-9-80-1-2 was purified on a Hydro column eluted with CH_3_OH−H_2_O (65:35) and CH_3_CN−H_2_O (40:60) to yield **9** (2.5 mg). Subfraction Pr-4718-9-80-1-3 was purified on a Hydro column with CH_3_CN−H_2_O−TFA (50:50:0.1) to yield **7** (4.5 mg). Subfraction 4718-9-80-1-10 was purified on a YMC NEA (R)-NP column eluted with CH_3_OH−H_2_O (40:60) to yield **10** (1.0 mg). Subfraction Pr-4718-9-80-1-I (164.3 mg) was purified on a Hydro column eluted with CH_3_OH−H_2_O−TFA (55:45:0.1) to yield **1** (4.0 mg) and **3** (6.4 mg). Compound **8** (19 mg) was obtained by the crystallization of a poorly soluble precipitate from fraction Pr-4718-9.

### 4.7. Spectral Data

Sajaroketide A (**1**): yellow amorphous; [α]_D_ ^20^ −1.38 (c 0.14 MeOH); UV (MeOH) λ_max_ (log ε) 205 (4.67), 247 (4.02), 240 (3.99), 337 (3.77) and 277 (3.48) nm; CD (c 0.00022 M, MeOH), λ_max_ (∆ε) 202 (−1.35), 218 (−1.37), 231 (−1.33), 263 (+1.47), 289 (+0.55), 300 (+0.67), 305 (+0.71), 307 (+0.70) and 340 (−0.52) nm; ^1^H and ^13^C NMR data, see Table 1, Appendix A; HRESIMS [M + Na]^+^ *m*/*z* 259.0576 (calcd. for C_12_H_12_O_5_Na 259.0577, ∆-0.4 ppm) (Appendix A).

Sajaroketide B (**2**): yellow amorphous; [α]_D_ ^20^ +3.12 (c 0.19 MeOH); UV (MeOH) λ_max_ (log ε) 275 (4.85), 238 (4.18), 247 (4.22), 338 (4.06) and 275 (3.56) nm; CD (c 0.00018 M, MeOH), λ_max_ (∆ε) 204 (−0.26), 219 (−1.32), 229 (−1.60), 262 (+1.91), 294 (+0.86), 340 (−0.22), 364 (−0.49) and 273 (−0.27) nm; ^1^H and ^13^C NMR data, see Table 1, Appendix A; HRESIMS [M + Na]^+^ *m*/*z* 273.0735 (calcd. for C_13_H_14_O_5_Na 373.0733, ∆-0.6 ppm) (Appendix A).

(2′S)-7-hydroxy-2-(2′-hydroxypropyl)-5-methylchromone (**3**): yellow amorphous; [α]_D_ ^20^ +21.3 (c 0.19 MeOH); UV (MeOH) λ_max_ (log ε) 213 (3.39), 250 (3.33), 197 (3.31), 202 (3.28), 238 (3.27) and 259 (2.67) nm; CD (c 0.00063 M, MeOH), λ_max_ (∆ε) 211 (-0.05), 224 (+0.58), 243 (+0.02), 250 (-1.12), 257 (+0.14), 273 (+0.32); 279 (+0.32), 289 (+0.32) nm; ^1^H NMR spectrum (500 MHz; acetone-d_6_; δ, ppm; *J* in Hz) 6.69 (1H, s, H-8), 6.66 (1H, s, H-6), 5.97 (1H, s, H-3), 4.20 (1H, m, H-2′), 2.71 (3H, s, H_3_-9), 2.66 (1H, dd, *J* = 14.5, 5.2, Ha-2′), 2.62 (1H, dd, *J* = 14.5, 5.2, Hb-2′), 1.25 (3H, d, *J* = 6.2, H_3_-3′) (Appendix A); ^13^C NMR spectrum (125 MHz; acetone-d_6_; δ, ppm) 179.4 (C-4), 165.5 (C-2), 161.5 (C-7), 160.6 (C-8a), 143.1 (C-5), 117.1 (C-6), 116.3 (C-4a), 112.8 (C-4), 101.6 (C-8), 65.5 (C-2′), 44.2 (C-1′), 23.7 (C-3′), 22.8 (C-9) (Appendix A); [M–H]^-^ *m*/*z* 233.0820 (calcd. for C_13_H_13_O_4_ 233.0819, ∆-0.4 ppm), HRESIMS [M + Na]^+^ *m*/*z* 257.0788 (calcd. for C_13_H_14_O_4_Na 357.0784, ∆-1.3 ppm (Appendix A).

Altechromone A (**4**): yellow amorphous; ^1^H NMR spectrum (500 MHz; acetone-d_6_; δ, ppm; *J* in Hz) 6.67 (1H, d, *J* = 2.2, H-8), 6.66 (1H, dd, *J* = 2.2, 0.7, H-6), 5.91 (1H, d, *J* = 0.7, H-3), 2.70 (3H, brs, H_3_-9), 2.28 (3H, d, *J* = 0.7, H_3_-10) (Appendix A); ^13^C NMR spectrum (125 MHz; acetone-d_6_; δ, ppm) 179.3 (C-4), 164.5 (C-2), 161.6 (C-7), 160.6 (C-8a), 143.1 (C-5), 117.1 (C-6), 116.0 (C-4a), 111.8 (C-4), 101.6 (C-8), 22.8 (C-9), 19.7 (C-10) (Appendix A); HRESIMS [M + Na]^+^ *m*/*z* 213.0522 (calcd. for C_11_H_10_O_3_Na 213.0522, ∆ 0.0 ppm; [M–H]^-^ *m*/*z* 189.0559 (calcd. for C_11_H_10_O_3_ 189.0557, ∆-1.1 ppm) (Appendix A).

Norlichexanthone (**5**): yellow amorphous; ^1^H NMR spectrum (500 MHz; acetone-d_6_; δ, ppm; *J* in Hz) 13.4 (1H, s, 1-OH), 6.70 (1H, brs, H-5), 6.70 (1H, brs, H-7), 6.30 (1H, d, *J* = 2.0, H-4), 6.19 (1H, d, *J* = 2.0, H-2), 2.78 (3H, s, 8-CH_3_) (Appendix A); ^13^C NMR spectrum (125 MHz; acetone-d_6_; δ, ppm) 181.7 (C-9), 164.2 (C-3), 163.6 (C-1), 162.2 (C-6), 158.9 (C-5a), 156.7 (C-4a), 143.1 (C-4a), 115.6 (C-7), 111.3 (C-8a), 102.5 (C-1a), 100.2 (C-5), 97.4 (C-2), 92.7 (C-4), 22.0 (8-CH_3_) (Appendix A); HRESIMS [M + Na]^+^ *m*/*z* 213.0522 (calcd. for C_11_H_10_O_3_Na 213.0522, ∆ 0.0 ppm; [M–H]^-^ *m*/*z* 189.0559 (calcd. for C_11_H_9_O_3_ 189.0557, ∆-1.1 ppm) (Appendix A).

Griseoxanthone C (**6**): orange amorphous; ^1^H NMR spectrum (300 MHz; acetone-d_6_; δ, ppm; *J* in Hz) 13.4 (1H, s, 1-OH), 6.71 (1H, brs, H-5), 6.71 (1H, brs, H-7), 6.40 (1H, d, *J* = 2.5, H-4), 6.25 (1H, d, *J* = 2.5, H-2), 2.78 (3H, s, 8-CH_3_) (Appendix A); ^13^C NMR spectrum (75 MHz; acetone-d_6_; δ, ppm) 181.7 (C-9), 167.1 (C-3), 164.7 (C-1), 163.6 (C-6), 160.4 (C-5a), 158.0 (C-4a), 144.6 (C-8), 117.0 (C-8a), 112.7 (C-7), 104.4 (C-1a), 101.6 (C-5), 97.6 (C-2), 92.6 (C-4), 56.3 (3-OCH_3_), 23.4 (8-CH_3_) (Appendix A); HRESIMS [M + Na]^+^ *m*/*z* 295.0575 (calcd. for C_15_H_12_O_5_Na 295.0577, ∆ 0.6 ppm; [M–H]^-^ *m*/*z* 271.0612 (calcd. for C_15_H_11_O_5_ 271.0612, ∆-0.1 ppm) (Appendix A).

1,3,5,6-Tetrahydroxy-8-methylxanthone (**7**): yellow amorphous ^1^H NMR spectrum (700 MHz; acetone-d_6_; δ, ppm; *J* in Hz) 13.4 (1H, s, 1-OH), 6.72 (1H, brs, H-7), 6.38 (1H, d, *J* = 2.2, H-4), 6.19 (1H, d, *J* = 2.2, H-2), 2.72 (3H, s, 8-CH_3_) (Appendix A); ^13^C NMR spectrum (175 MHz; acetone-d_6_; δ, ppm) 183.5 (C-9), 165.0 (C-3), 164.4 (C-1), 157.9 (C-4a), 150.9 (C-6), 148.0 (C-5a), 133.1 (C-8), 131.4 (C-5), 116.1 (C-7), 112.8 (C-8a), 103.6 (C-1a), 98.7 (C-2), 94.2 (C-4), 22.7 (8-CH_3_) (Appendix A); HRESIMS [M + Na]^+^ *m*/*z* 297.0372 (calcd. for C_14_H_10_O_6_Na 297.0370, ∆ -0.7 ppm; [M–H]^-^ *m*/*z* 273.0409 (calcd. for C_15_H_9_O_6_ 273.0405, ∆-1.5 ppm); [M + H]^+^ *m*/*z* 275.0553 (calcd. for C_15_H_11_O_6_ 275.0550, ∆-0.9 ppm) (Appendix A).

Griseofulvin (**8**): colorless powder; ^1^H NMR spectrum (700 MHz; CDCl_3_; δ, ppm; *J* in Hz) 6.12 (1H, s, H-5), 5.53 (1H, s, H-3′), 4.02 (3H, s, 6-OCH_3_), 3.97 (3H, s, 4-OCH_3_), 3.60 (3H, s, 2′-OCH_3_), 3.01 (1H, dd, *J* = 16.8, 13.5, Ha-5′), 2.82 (1H, m, H-6′), 2.42 (1H, dd, *J* = 16.8, 4.7, Hb-5′), 0.95 (3H, d, *J*= 6.7, H-7′) (Appendix A); ^13^C NMR spectrum (175 MHz; CDCl_3_; δ, ppm) 197.1 (C-4′), 192.6 (C-3), 170.9 (C-2′), 169.7 (C-7a), 164.8 (C-6), 157.9 (C-4), 105.3 (C-3a), 105.0 (C-3′), 97.5 (C-7), 90.9 (C-1′), 89.7 (C-5), 57.1 (6-OCH_3_), 56.8 (2′-OCH_3_), 56.5 (4-OCH_3_), 40.2 (C-5′), 36.6 (C-6′), 14.4 (C-7′) (Appendix A); HRESIMS [M + Na]^+^ *m*/*z* 375.0613 (calcd. for C_17_H_17_ClO_6_Na 375.0606, ∆ -1.8 ppm; [M + H]^+^ *m*/*z* 353.0793 (calcd. for C_17_H_18_ClO_6_ 353.0786, ∆-2.0 ppm) (Appendix A).

6-O-Desmethylgriseofulvin (**9**): yellow amorphous; ^1^H NMR spectrum (500 MHz; acetone-d_6_; δ, ppm; *J* in Hz) 6.50 (1H, s, H-5), 5.57 (1H, s, H-3′), 3.90 (3H, s, 4-OCH_3_), 3.72 (3H, s, 2′-OCH_3_), 2.88 (1H, m, Ha-5′), 2.85 (1H, m, H-6′), 2.39 (1H, d, *J* = 12.4, Hb-5′), 0.96 (3H, d, *J* = 6.7, H-7′) (Appendix A); ^13^C NMR spectrum (125 MHz; acetone-d_6_; δ, ppm) 196.8 (C-4′), 192.7 (C-3), 172.4 (C-2′), 171.9 (C-7a), 165.5 (C-6), 159.4 (C-4), 106.0 (C-3′), 105.9 (C-3a), 96.8 (C-7), 95.7 (C-5), 92.1 (C-1′), 58.0 (2′-OCH_3_), 57.3 (4-OCH_3_), 41.4 (C-5′), 37.9 (C-6′), 15.2 (C-7′) (Appendix A); HRESIMS [M + Na]^+^ *m*/*z* 362.0450 (calcd. for C_16_H_15_ClO_6_Na 361.0449, ∆ -0.3 ppm; [M–H]^-^ *m*/*z* 337.0483 (calcd. for C_16_H_14_ClO_6_ 337.0484, ∆ 0.3 ppm) (Appendix A).

Deschlorogriseofulvin (**10**): yellow amorphous; ^1^H NMR spectrum (500 MHz; acetone-d_6_; δ, ppm; *J* in Hz) 6.24 (1H, d, *J* = 1.9, H-5), 6.39 (1H, d, *J* = 1.9, H-7), 5.54 (3H, brs, H-3′), 3.99 (3H, s, 6-OCH_3_), 3.93 (3H, s, 4-OCH_3_), 3.71 (3H, s, 2′-OCH_3_), 2.89 (1H, m, Ha-5′), 2.77 (1H, m, H-6′), 2.34 (1H, dd, *J* = 16.3, 4.6, Hb-5′), 0.94 (3H, d, *J* = 6.6, H-7′) (Appendix A); ^13^C NMR spectrum (125 MHz; acetone-d_6_; δ, ppm) 196.8 (C-4′), 193.0 (C-3), 177.6 (C-7a), 172.8 (C-2′), 172.1 (C-6), 160.8 (C-4), 105.8 (C-3′), 105.7 (C-3a), 94.7 (C-7), 91.2 (C-1′), 90.4 (C-5), 57.8 (2′-OCH_3_), 57.5 (6-OCH_3_), 57.1 (4-OCH_3_), 41.4 (C-5′), 38.0 (C-6′), 15.3 (C-7′) (Appendix A); HRESIMS [M + Na]^+^ *m*/*z* 341.1008 (calcd. for C_17_H_18_O_6_Na 341.0996, ∆ -3.5 ppm; [M + H]^+^ *m*/*z* 319.1187 (calcd. for C_17_H_19_O_6_ 319.1176, ∆-3.4 ppm) (Appendix A).

5,6-Dihydro-4-methyl-2H-pyran-2-one (**11**): colorless amorphous; ^1^H NMR spectrum (500 MHz; acetone-d_6_; δ, ppm; *J* in Hz) 5.70 (1H, q, *J* = 1.5, H-3), 4.32 (1H, d, *J* = 6.3, Ha-6), 4.31 (1H, d, *J* = 6.3, Hb-6), 2.42 (1H, d, *J* = 6.5, Ha-5), 2.41 (1H, d, *J* = 6.5, Hb-5), 1.99 (3H, brs, H_3_-7) (Appendix A); ^13^C NMR spectrum (125 MHz; acetone-d_6_; δ, ppm) 164.5 (C-2), 159.4 (C-4), 66.5 (C-6), 28.7 (C-5), 22.7 (C-4) (Appendix A); HRESIMS [M + Na]^+^ *m*/*z* 135.0413 (calcd. for C_6_H_8_O_2_Na 135.0417, ∆ 2.8 ppm) (Appendix A).

### 4.8. Quantum-Chemical Modeling

The quantum-chemical modeling of the geometry and spectroscopic properties of compounds **1** and **2** were performed using the Gaussian 16 package of programs [20]. Geometry optimizations and calculations of the IR spectra were conducted with B3LYP exchange-correlation functional, the polarization continuum model (PCM), and 6-311+G(d,p) and cc-pvTz split-valence basis sets. 

The statistical weights (gim) of conformations were calculated via Gibbs free energies:(1)gim=e−ΔGim/RT∑ie−ΔGim/RT
where index “m” denotes the most stable conformation and ΔGim = Gi − Gm are the relative Gibbs free energies. 

The ECD spectra were calculated using the time-dependent density functional theory (TDDFT), B3LYP functional, PCM model, and cc-pvTz basis set. Thirty electronic transitions were calculated for each conformation of 1 and 2. The individual bands in the theoretical spectra were simulated as a Gauss-type functions with the bandwidth ζ = 0.28 eV. The UV shift Δλ = +7 nm was used for the best correspondence between the experimental and calculated spectra for **1** and **2**.

### 4.9. Urease Inhibition Assay

A reaction mixture consisting of 25 µL of enzyme solution (urease from *Canavalia ensiformis*, Sigma, 1U final concentration) and 5 µL of test compounds dissolved in water (10–300.0 µM final concentration) was preincubated at 37 °C for 60 min in 96-well plates. Then, 55 µL of phosphate-buffered solution with 100 µM of urea was added to each well and incubated at 37 °C for 10 min. The urease inhibitory activity was estimated by determining ammonia production using the indophenol method. Briefly, 45 µL of phenol reagent (1% *w*/*v* phenol and 0.005% *w*/*v* sodium nitroprusside) and 70 µL of alkali reagent (0.5% *w*/*v* NaOH and 0.1% active chloride NaClO) were added to each well. The absorbance was measured after 50 min at 630 nm using a MultiskanFS microplate reader (Thermo Scientific Inc., Beverly, MA, USA). All reactions were performed in triplicate in a final volume of 200 µL. The pH was maintained at 7.3–7.5 in all assays. DMSO 5% was used as a positive control.

### 4.10. Antimicrobial Activity

The yeast-like fungi of *Candida albicans* KMM 455 and bacterial strains *Staphylococcus aureus* ATCC 21027 and *Escherichia coli* VKPM (B-7935) (Collection of Marine Microorganisms PIBOC FEB RAS) were cultured on solid-medium Mueller Hinton broth with agar (16.0 g/L) in a Petri dish at 37 °C for 24 h. 

The assays were performed in 96-well microplates in appropriate Mueller Hinton broth. Each well contained 90 µL of bacterial or of a yeast-like fungi suspension (10^9^ CFU/mL). Then, 10 µL of a compound diluted at concentrations ranging from 1.5 to 100.0 µM using a 2-fold dilution was added (DMSO concentration < 1%). Culture plates were incubated overnight at 37 °C, and the OD_620_ was measured using a Multiskan FS spectrophotometer (Thermo Scientific Inc., Beverly, MA, USA). The antibiotic gentamicin and antifungal agent nitrofungin were used as positive controls at 1 mg/mL; 1% DMSO in PBS served as a negative control. 

### 4.11. Cell Culture

The rat cardiomyocyte H9c2 cells were kindly provided by Prof. Dr. Gunhild von Amsberg from the Martini-Klinik Prostate Cancer Center, University Hospital Hamburg-Eppendorf, Hamburg, Germany. The human hepatocarcinoma cell HepG2 was obtained from ATCC.

The HepG2 and H9c2 cells were cultured in DMEM medium (Biolot, St. Petersburg, Russia) containing 10% fetal bovine serum (Biolot, St. Petersburg, Russia) and 1% penicillin/streptomycin (Biolot, St. Petersburg, Russia) at 37 °C in a humidified atmosphere with 5% (*v*/*v*) CO_2_. 

### 4.12. Cell Viability Assay

The HepG2 and H9c2 cells were seeded at concentrations of 5× 10^3^ and 3×10^3^ cell/well, respectively, and the experiments were started after 24 h. The compounds at concentrations up to 100 µM were added into the wells for 24 h, and the viability of the cells was measured by an MTT (3-(4,5-dimethylthiazol-2-yl)-2,5-diphenyltetrazolium bromide) assay, which was performed according to the manufacturer’s instructions (Sigma-Aldrich, Munich, Germany). All compounds were dissolved with DMSO so that the final concentration of DMSO in the cell culture was not more than 1%. Moreover, DMSO was used as a control. The absorbance of the resulting solution was measured at 570 nm with a microplate reader MultiscanFC (ThermoLabsystems Inc., Beverly, MA, USA). The results are presented as a percent of the control data and calculated IC_50_.

### 4.13. Staphylococcus aureus-Induced Infection of H9c2 Cardiomyocytes

The H9c2 cells were seeded at concentrations of 3 × 10^3^ cell/well. After 24 h, the cell culture media was changed with an *S. aureus* bacterial suspension (10^2^ CFU/mL) prepared in full DMEM culture media. The investigated compounds at concentrations of 1 and 10 µM were added after 1 h and the cells were incubated for 48 h. The *S. aureus* suspension with compounds at the same concentrations was used as a control. MTT reagent was added to all wells and formazan production was detected with the microplate reader MultiscanFC (ThermoLabsystems Inc., Beverly, MA, USA). The viability of *S. aureus*-infected cells was calculated as:Viability, relative units = (OD cells+Sa +comp − OD Sa+comp)/(OD cells+Sa − OD Sa)

### 4.14. Statistical Data Evaluation

All the data were obtained in three independent replicates, and the calculated values are expressed as a mean ± standard error mean (SEM). Student’s t-test was performed using SigmaPlot 14.0 (Systat Software Inc., San Jose, CA, USA) to determine the statistical significance. Differences were considered statistically significant at *p* < 0.05.

## 5. Conclusions

Thus, we reported the isolation and structure elucidation of two new polyketides, sajaroketides A (1) and B (2), together with known (2′S)-7-hydroxy-2-(2′-hydroxypropyl)-5-methylchromone (3), altechromone A (4), norlichexanthone (fusarindin) (5), griseoxanthone C (6), 1,3,5,6-tetrahydroxy-8-methylxanthone (7), griseofulvin (8), 6-O-desmethylgriseofulvin (9), dechlorogriseofulvin (10), and 5,6-dihydro-4-methyl-2H-pyran-2-one (11) from the natural complex of the marine-derived fungal strains *Penicillium sajarovii* KMM 4718 and *Aspergillus protuberus* KMM 4747 isolated from the sea urchin *Scaphechinus mirabilis* (Sea of Japan). The fungal strains *P. sajarovii* KMM 4718 and *A. protuberus* KMM 4747 were identified as and based on three molecular markers, *ITS*, *BenA*, *CaM* regions, as well as the *RPB2* region for KMM 4747. The absolute configurations of the new compounds 1 and 2 were elucidated by quantum chemical calculations. The effects of these compounds on urease activity and the growth of *Staphylococcus aureus*, *Escherichia coli*, and *Candida albicans* were found. Sajaroketide A, altechromone A, and griseofulvin showed significant cardioprotective effects in an in vitro model of *S. aureus*-induced infectious myocarditis. 

## Figures and Tables

**Figure 1 ijms-24-16568-f001:**
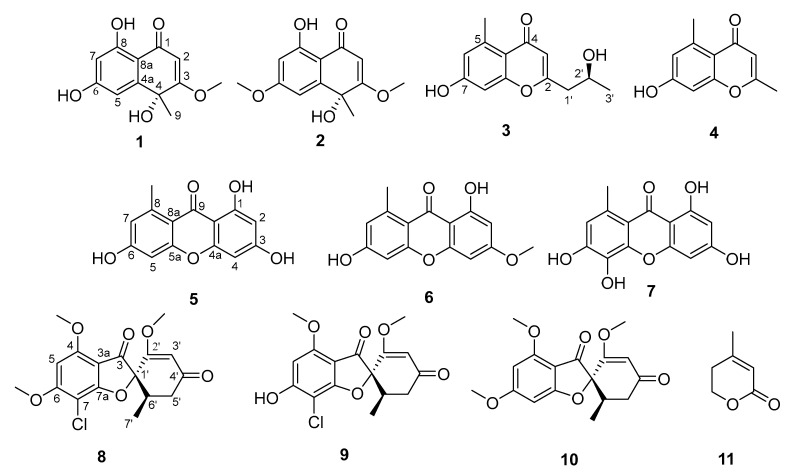
Metabolites isolated from the natural complex of the fungi *P. sajarovii* KMM 4718 and *A. protuberus* KMM 4747 isolated from the sea urchin *S. mirabilis*.

**Figure 2 ijms-24-16568-f002:**
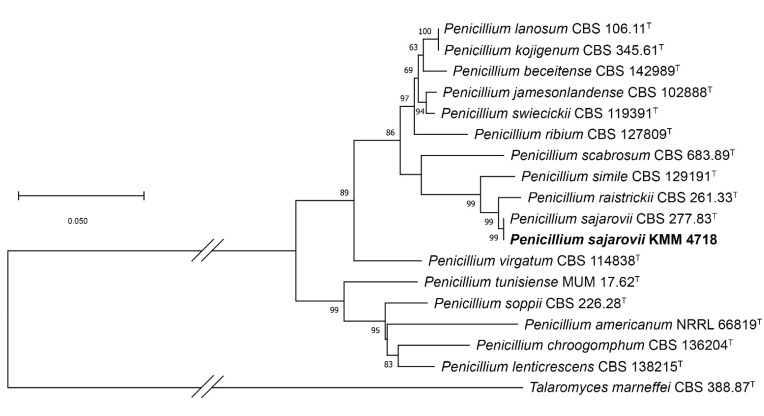
ML tree based on concatenated *ITS*-*BenA*-*CaM* gene sequences showing the phylogenetic position of strain KMM 4718 among members of the genus *Penicillium* (*Ramosum*), series *Lanosa*, *Raistrickiorum*, *Scabrosa*, *Soppiorum*, and *Virgata*. Bootstrap values (%) of 1000 replications. Nodes with confidence values higher than 50% are indicated. The scale bars represent 0.05 substitutions per site.

**Figure 3 ijms-24-16568-f003:**
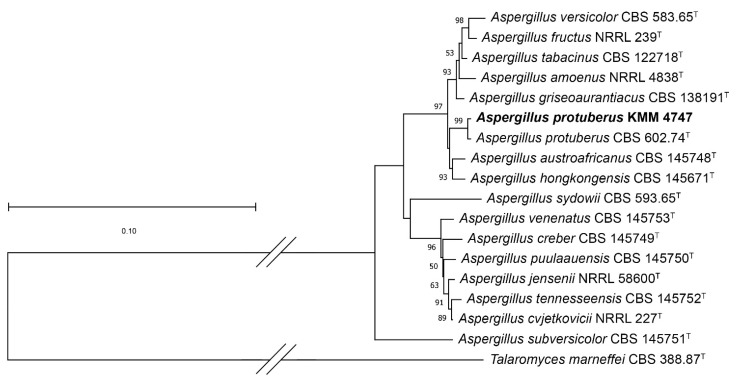
ML tree based on concatenated *ITS*-*BenA*-*CaM-RPB2* gene sequences showing the phylogenetic position of strain KMM 4747 among the members of the genus *Aspergillus* series *Versicolores*. Bootstrap values (%) of 1000 replications. Nodes with confidence values higher than 50% are indicated. The scale bars represent 0.1 substitutions per site.

**Figure 4 ijms-24-16568-f004:**
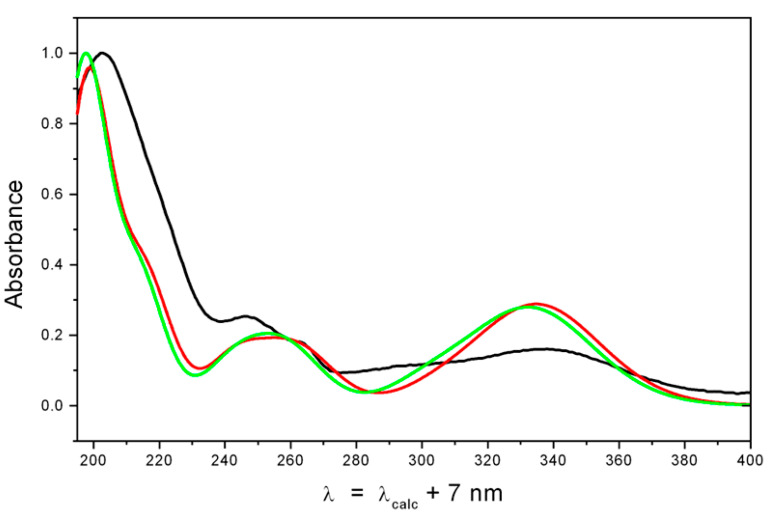
UV spectra of **1**: experimental (black), calculated for **4*R*-1_c1** (red), and calculated for **4*R*-1_c2** (green). Theoretical spectra are calculated at the “PCM level” of theory.

**Figure 5 ijms-24-16568-f005:**
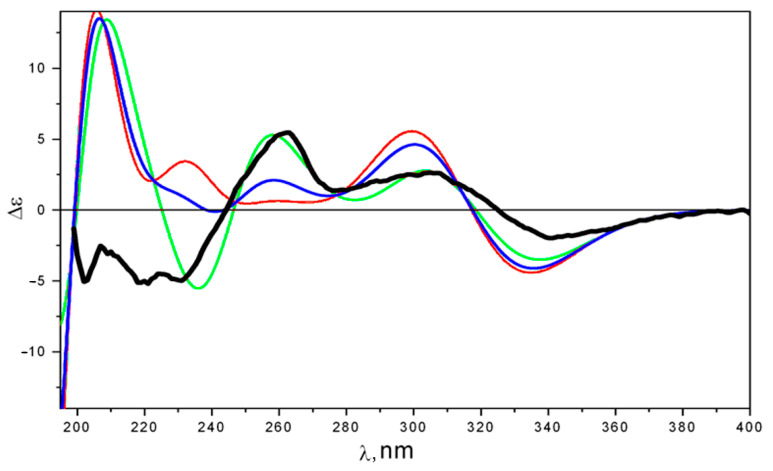
Experimental (black) and theoretical ECD spectra, calculated for 4*R*-**1** at the TDDFT_B3LYP/cc-pvTZ_PCM//B3LYP/cc-pvTZ_PCM level of theory: calculated for **4*R*-1_c1** (red), **4*R*-1_c2** (green), and calculated averaged for 4*R*-**1** (blue).

**Figure 6 ijms-24-16568-f006:**
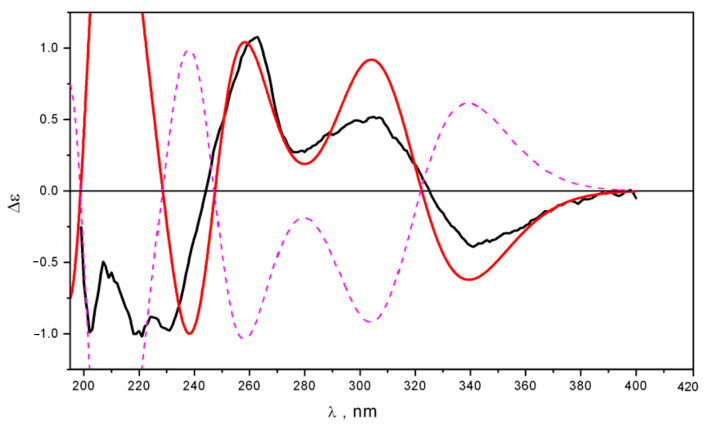
Theoretical averaged ECD spectrum, calculated for 4*R*-**1** (red) and 4*S*-**1** (pink) using “Direct” modeling, compared with the experimental ECD spectrum of **1** (black).

**Figure 7 ijms-24-16568-f007:**
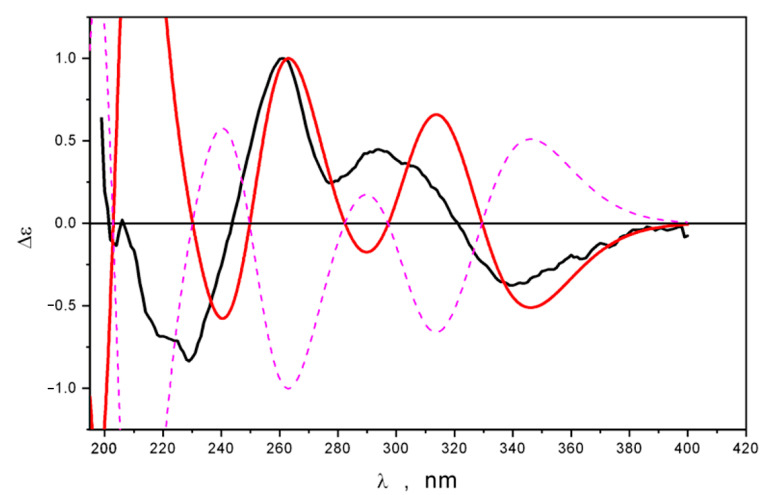
Theoretical averaged ECD spectrum, calculated for 4*R*-**2** (red) and 4*S*-**2** (pink) using “Direct” modeling, compared with the experimental ECD spectrum of **2** (black).

**Figure 8 ijms-24-16568-f008:**
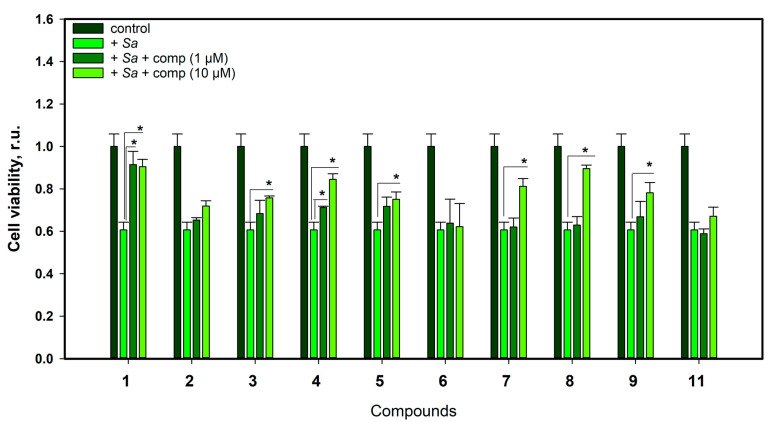
The viability of *S. aureus*-infected H9c2 cardiomyocytes treated with compounds **1**–**9** and **11**. The compounds at concentrations of 1 and 10 µM were added after 1 h of *S. aureus* infection. All the data are presented as means ± standard errors of means. The experiments are conducted in three independent replicates. Asterisk * indicates the significant differences at *p* ≤ 0.05.

**Table 1 ijms-24-16568-t001:** ^13^C and ^1^H NMR spectroscopic data for compounds **1**–**2**.

No	1		2	
	δ_C_, Type	δ_H_, Mult. (*J* in Hz)	δ_C_, Type	δ_H_, Mult. (*J* in Hz)
1	190.3, C		190.4, C	
2	99.6, CH	5.52, s	99.6, CH	5.55, s
3	179.3, C		179.6, C	
4	70.7, C		70.9, C	
4a	151.5, C		151.1, C	
5	106.1, CH	6.77, d (2.5)	105.2, CH	6.81, d (2.5)
6	164.1, C		166.1, C	
7	102.4, CH	6.25, d (2.5)	100.6, CH	6.35, d (2.5)
8	164.4, C		164.7, C	
8a	107.5, C		108.0, C	
9	32.5, CH_3_	1.60, s	32.6, CH_3_	1.62, s
3-OMe	56.9, CH_3_	3.89, s	57.0, CH_3_	3.90, s
6-OMe			55.9, CH_3_	3.87, s
8-OH				13.2, s

Chemical shifts are measured at 125 and 500 MHz in acetone-d_6._

**Table 2 ijms-24-16568-t002:** Inhibition of urease and antimicrobial activities of the compounds.

Compounds	Inhibition of Urease Activity IC_50_, µM	Inhibition of Microbial Growth ^1^, %		
		*S. aureus*	*E. coli*	*C. albicans*
**1**	>100.0	35.1 ± 0.9	25.4 ± 1.0	-
**2**	98.5 ± 0.8	-	-	-
**3**	>100.0	17.7 ± 0.3	-	-
**4**	>100.0	30.1 ± 1.2	13.6 ± 0.3	11.1 ± 1.3
**5**	15.3 ± 2.1	20.8 ± 6.5	27.9 ± 3.3	-
**6**	97.3 ± 1.6	-	-	-
**7**	100.0	-	-	15.4 ± 2.1
**8**	10.1 ± 0.6	11.5 ± 4.3	16.8 ± 4.5	-
**9**	>100.0	23.0 ± 0.3	23.4 ± 1.8	-
**11**	11.4 ± 1.0	-	-	-

^1^ Compounds are tested at a concentration of 100 µM.

**Table 3 ijms-24-16568-t003:** Cytotoxic activities of the compounds.

Compound	HepG2			H9c2		
	Cell Viability, %		IC_50_	Cell Viability, %		IC_50_
	100 µM	10 µM		100 µM	10 µM	
**1**	106.8 ± 3.9	107.5 ± 3.2	>100	65.4 ± 4.8	83.6 ± 4.4	>100
**2**	76.9 ± 2.1	87.3 ± 4.1	>100	79.0 ± 3.4	96.0 ± 1.9	>100
**3**	101.4 ± 3.9	91.7 ± 1.5	>100	74.3 ± 3.2	96.4 ± 2.1	>100
**4**	92.6 ± 4.1	99.2 ± 6.5	>100	96.5 ± 3.6	110.3 ± 2.9	>100
**5**	52.6 ± 3.0	88.0 ± 4.9	>100	49.0 ± 1.6	86.3 ± 6.3	97.4 ± 2.2
**6**	27.1 ± 2.4	85.7 ± 5.7	64.7 ± 0.8	63.3 ± 1.4	86.0 ± 4.8	>100
**7**	57.4 ± 4.2	77.1 ± 1.4	>100	52.9 ± 1.5	98.0 ± 0.9	>100
**8**	39.9 ± 0.7	81.0 ± 2.1	77.8 ± 1.3	71.6 ± 3.7	89.8 ± 2.7	>100
**9**	84.2 ± 3.2	91.5 ± 6.8	>100	92.9 ± 3.7	97.4 ± 8.1	>100
**11**	76.6 ± 2.4	87.5 ± 5.3	>100	89.0 ± 2.9	95.9 ± 0.6	>100

**Table 4 ijms-24-16568-t004:** The strains of the species used in the multi-locus phylogenetic analysis and GenBank accession numbers.

Species	Strain Number	GenBank Accession Number			
		*ITS*	*BenA*	*CaM*	*RPB2*
*Penicillium simile* Davolos et al. Davolos et al., 2012	CBS 129191 ^T^	FJ376592	FJ376595	GQ979710	–
*Penicillium sajarovii* Quintan	CBS 277.83 ^T^	KC411724	MN969397	MN969295	–
*Penicillium sajarovii*	KMM 4718	OR431843	OQ466617	OR451997	–
*Penicillium raistrickii* G. Sm	CBS 261.33 ^T^	AY373927	KJ834485	KJ867006	–
*Penicillium lanosum* Westling	CBS 106.11 ^T^	DQ304540	DQ285627	FJ530974	–
*Penicillium beceitense* Guevara-Suarez et al.	CBS 142989 ^T^	LT899780	LT898229	LT899764	–
*Penicillium jamesonlandense* Frisvad & Overy	CBS 102888 ^T^	DQ267912	DQ309448	KJ866985	–
*Penicillium swiecickii* Zaleski	CBS 119391 ^T^	AF033490	KJ834494	KJ866993	–
*Penicillium ribium* Frisvad & Overy	CBS 127809 ^T^	DQ267916	MN969395	KJ866995	–
*Penicillium scabrosum* Frisvad et al.	CBS 683.89 ^T^	DQ267906	DQ285610	FJ530987	–
*Penicillium chroogomphum* Xu et al.	CBS 136204 ^T^	KC594043	KP684056	KP684057	–
*Penicillium americanum* Jurjevic et al.	NRRL 66819 ^T^	MK791278	MK803427	MK803428	–
*Penicillium lenticrescens* Visagie et al.	CBS 138215 ^T^	KJ775675	KJ775168	KJ775404	–
*Penicillium soppii* Zaleski	CBS 226.28 ^T^	AF033488	MN969399	KJ867002	–
*Penicillium tunisiense* Ouhibi et al.	MUM 17.62 ^T^	MG586956	MG586970	MG586974	–
*Penicillium virgatum* Nirenberg & Kwasna	CBS 114838 ^T^	AJ748692	KJ834500	KJ866992	–
*Penicillium kojigenum* Sm	CBS 345.61 ^T^	AF033489	KJ834463	KJ867011	–
*Aspergillus versicolor* (Vuill.) Tirab	CBS 583.65 ^T^	EF652442	EF652266	EF652354	EF652178
*Aspergillus fructus* Jurjevic et al.	NRRL 239 ^T^	EF652449	EF652273	EF652361	EF652185
*Aspergillus tabacinus* Nakaz et al.	CBS 122718 ^T^	EF652478	EF652302	EF652390	EF652214
*Aspergillus amoenus* Roberg	NRRL 4838 ^T^	EF652480	JN853946	JN854035	JN853824
*Aspergillus griseoaurantiacus* Visagie et al.	CBS 138191 ^T^	KJ775553	KJ775086	KJ775357	KU866988
*Aspergillus austroafricanus* Jurjevic et al.	CBS 145748 ^T^	JQ301891	JN853963	JN854025	JN853814
*Aspergillus hongkongensis* Tsang et al.	CBS 145671 ^T^	AB987907	LC000552	MN969320	LC000578
*Aspergillus protuberus* Munt.-Cvetk	CBS 602.74 ^T^	EF652460	EF652284	EF652372	EF652196
*Aspergillus protuberus*	KMM 4747	OR431842	OQ466615	OR451996	OR451998
*Aspergillus tennesseensis* Jurjevic et al.	CBS 145752 ^T^	JQ301895	JN853976	JN854017	JN853806
*Aspergillus cvjetkovicii* Jurjevic et al.	NRRL 227 ^T^	EF652440	EF652264	EF652352	EF652176
*Aspergillus jensenii* Jurjevic et al.	NRRL 58600 ^T^	JQ301892	JN854007	JN854046	JN853835
*Aspergillus puulaauensis* Jurjevic et al.	CBS 145750 ^T^	JQ301893	JN853979	JN854034	JN853823
*Aspergillus venenatus* Jurjevic et al.	CBS 145753 ^T^	JQ301896	JN854003	JN854014	JN853803
*Aspergillus creber* Jurjevic et al.	CBS 145749 ^T^	JQ301889	JN853980	JN854043	JN853832
*Aspergillus sydowii* (Bainier & Sartory) Thom & Church	CBS 593.65 ^T^	EF652450	EF652274	EF652362	EF652186
*Aspergillus subversicolor* Jurjevic et al.	CBS 145751 ^T^	JQ301894	JN853970	JN854010	JN853799
*Talaromyces marneffei* (Segretain) Samson et al.	CBS 388.87 ^T^	JN899344	JX091389	KF741958	KM023283

Note: ex-type strains *Penicillium lusitanum* MUM 18.49 (the sequence for the *CaM* gene is not established) and *Aspergillus pepii* CBS 142028 (the sequence for the *RPB2* gene i not been established) are not used for the phylogenetic analysis.

## Data Availability

The original data presented in the study are included in the article/Appendix A; further inquiries can be directed to the corresponding authors.

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
