# Peer review of "Bioactive Polyketides from the Natural Complex of the Sea Urchin-Associated Fungi Penicillium sajarovii KMM 4718 and Aspergillus protuberus KMM 4747"

_ijms, 2023, doi:10.3390/ijms242316568_

Round 1

Reviewer 1 Report

Comments and Suggestions for Authors

The authors isolated two fungal strains from the sea urchin Scaphechinus mirabilis. Two new and 9 known natural products were identified. The structures of compounds 1 and 2 were elucidated by HRMS and detailed NMR spectroscopy. The absolute configuration of new compounds was established by ECD spectra and quantum chemical calculations. The urease inhibition and antimicrobial activity was studied too.

According to the database SciFinder compounds 1 and 2 are new.

The manuscript is suitable for publication after minor revision.

Line 51: our study (instead of investigation).

Figure 1 and 3 are unreadable.

The doi numbers are missing from the references.

References [8, 9, 11]: the monograms of authors are given only.

Comments on the Quality of English Language

Line 51: our study (instead of investigation).

Author Response

Reviewer 1:

We thank the reviewer for the high appreciation of our efforts.

Q: Line 51: our study (instead of investigation).

 A: We thank the reviewer for the for valuable remark. Accordingly, we did these changes.

Q: Figure 1 and 3 are unreadable.

A: We thank the reviewer for the for valuable remark. We add clear Figures 1-3 in manuscript, and add Figures 2 and 3 in .png for editorial office

Q: The doi numbers are missing from the references.

A: We thank the reviewer for the for valuable remark. We checked all references and add doi

Q:  References [8, 9, 11]: the monograms of authors are given only.

A: We thank the reviewer for the for valuable remark. We checked all references

Reviewer 2 Report

Comments and Suggestions for Authors

The authors describe the isolation and characterization of two new compounds. The concrete ones are numbered 1 and 2. Their structural elucidation is correct, and based on mass spectrometry and one- and two-dimensional nuclear magnetic resonance.

I would only like to indicate that in the spectra attached in the supporting information, it would be convenient to include the chemical structure of the compound in question, for a better identification of the signals.

Author Response

Reviewer 2:

We thank the reviewer for the high appreciation of our efforts.

Q: I would only like to indicate that in the spectra attached in the supporting information, it would be convenient to include the chemical structure of the compound in question, for a better identification of the signals.

 A: We thank the reviewer for the for valuable remark. Accordingly, we did these changes in SI.